# Reliability of Repeated Measures of Nutrient Intake by Diet Records in Residents in the Western Region of Japan

**DOI:** 10.3390/nu11102515

**Published:** 2019-10-18

**Authors:** Kazuko Yoshizawa, Walter C. Willett, Changzheng Yuan

**Affiliations:** 1Department of Nutrition, Harvard T.H. Chan School of Public Health, Boston, MA 02115, USA; wwillett@hsph.harvard.edu (W.C.W.); chy478@mail.harvard.edu (C.Y.); 2Department of Epidemiology, Harvard T.H Chan School of Public Health, Boston, MA 02115, USA; 3Channing Division of Network Medicine, Brigham and Women’s Hospital, Harvard Medical School, Boston, MA 02115, USA; 4School of Public Health, Zhejiang University, Hangzhou 310027, China

**Keywords:** Intraclass Correlation Coefficient (ICC), reliability, energy adjustment, diet record within-person and between-person variation

## Abstract

Objective: We aimed to assess the day-to-day variation in twelve one-day diet records over one year from 131 residents of urban and rural areas in the western region of Japan. Methods: Between 2014 and 2015, the participants provided repeated one-day diet records once a month. We estimated the intraclass correlation coefficient (ICCs) for intakes for energy and 39 crude and energy-adjusted nutrients using linear mixed models. Results: Among the unadjusted nutrients, ICCs ranged from 0.05 (95 percent confidence interval = 0.03–0.09) for vitamin A retinol equivalent (RE) to 0.55 (95% CI = 0.48–0.62) for potassium. After energy adjustment, the ICCs were 0.02 (95 percent confidence interval = 0.03–0.09) for vitamin A (RE) and 0.52 (95 percent confidence interval = 0.45–0.59) for potassium. Intakes of energy-adjusted macronutrients tended to have moderate degrees of day-to-day coefficients of variation (CV_w,_ range = 0.13–0.23, mean = 0.18), while the coefficients of variation for intakes of micronutrients varied dramatically (CV_w,_ range = 0.17–2.59, mean = 0.54). Conclusion: There were large day-to-day variations in nutrient intake assessed by diet records among urban and rural residents in the western region of Japan. This study provided information on the reproducibility of crude and energy-adjusted nutrients that may be useful for other dietary studies in Japanese populations.

## 1. Introduction

To assess relationships between dietary intake and health outcomes such as cardiovascular disease or cancer, information on a habitual diet is needed to represent long-term exposure. A critical feature of the diets of free-living individuals is variation from day to day. Understanding the daily dietary variation is essential for designing studies and interpreting the results in nutritional epidemiology. The magnitude of day-to-day variation differs greatly among nutrients and sometimes by the day of the week or season [1]. In a classical paper, Beaton and others evaluated factors contributing to the overall variation in nutrients within a population and concluded that random within-person and between-person variance were most important. Different interviewers and sequence of days of data collection made negligible contributions [2]. The degree of within-person variation in nutrient intake is partly cultural and is also dependent on food availability [1,2].

In nutritional epidemiology, we are usually interested in nutrients adjusted for total energy intake because individuals must make changes in nutrient intake primarily by changing the composition of their diet rather than by changing total energy intake. After energy adjustment, the proportion of within-person variation is usually substantially higher than that of between-person variation for many nutrients [1]. Intakes of macronutrients tend to have moderate degrees of day-to-day variation, but intakes of micronutrients can vary dramatically [1,3].

In epidemiological studies, the intraclass correlation coefficient (ICC) or reliability coefficient has been used to assess the reproducibility of dietary variables and biomarkers over time [4]. Studies evaluating the reproducibility of specific nutrients assessed by 24-hour recalls or diet records have been mainly conducted in western countries, and have documented the within- and between-person variation of many nutrients [1,5,6,7,8,9,10,11]. A few studies from Japan [12,13] reported within- and between-person components of variation for a limited number of nutrients without considering energy-adjusted values. Moreover, ICC, the direct measure for reliability, was not estimated in their studies.

We aimed to describe the crude (unadjusted for energy intake) and energy-adjusted ICCs, within- and between-person components of variation, and number of days required to estimate the true intakes to lie within the specified percentage of true means for energy and 39 nutrients obtained from twelve one-day diet records collected over one year from 131 residents of urban and rural areas in western Japan.

## 2. Methods

### 2.1. Participants in This Study

We planned to recruit study participants residing in urban and rural communities to capture geographical and cultural characteristics specific to the regions in Nagasaki Prefecture in the western part of Japan. Before entry into this study, they had lived in these areas for more than five years and had not recently changed their diet due to health. To achieve high compliance, we targeted highly motivated participants. We contacted local government health centers, health offices, and the board of directors of a nutrition volunteer organization at a prefectural level, and requested that they connect us with the residents, including nutrition volunteers at community levels. We also advertised targeting residents by a local government monthly magazine [14]. We sent interested participants invitations by mail and explained the objective and the methods of the study. In the end, 133 participants agreed to enter this study: 63% were community nutrition volunteers, 20% were municipal government employees, and 17% were members of the volunteers’ families or others. We started data collection from 133 participants, but two participants dropped out at the second diet survey due to their irregular work shifts. One hundred and thirty participants completed all 12 diet records and one completed only 11 records.

### 2.2. Collection of Diet Records and Computation of Nutrients Intakes

To incorporate variation due to seasonality and day of the week, we collected the one-day dietary information from the participants for 12 times over a one-year period, once a month, at approximately four-week intervals on different days of the week, between January 2014 and April 2015. The participants were asked to recall or record the names and portion sizes of all food and beverages consumed within 24-hours for breakfast, lunch, dinner, and snacks. We used the 24-hour recall at the beginning of the diet survey period, and then switched to the diet records by fax when the participants were able to make records. When the participants submitted the records to the dietitians by fax on the specified day, the dietitians confirmed missing information immediately after receiving the records. We continued to collect data by face-to-face interviews when necessary to assure accuracy.

The diet records were submitted each month to the headquarters after review by the district dietitians. The headquarters’ dietitians reviewed again all the diet records each month and prepared final records obtained from either the diet records or the 24-hour recalls. The final records included detailed information, including leftovers, necessary for estimating food intakes and calculating nutrient intakes. Early in the diet survey, the dietitians instructed the participants on how to conduct the diet record using a simple form prepared for this study. The participants were encouraged to use a scale when it was difficult for them to quantify their consumption level.

We computed nutrient intakes from the diet records by the use of software (*Excel Add-in Software Ver. 6.0*; Kenpakusha) with The Standard Tables of Food Composition Japan 2010 [15] and Fatty Acid Composition of Foods 2005 [16]. Our computation method was similar to the method used in Japan’s National Health and Nutrition Surveys (NHNS) [17,18], except that the national survey estimates individual consumption proportionally from the consumption of the entire family. In our study, we computed the amount of all consumed food and beverages on an individual basis. We found some food and beverages consumed were not listed on the composition tables. In this case, we used the values for similar foods. We did not include supplements, except for dried green vegetable powders (5 g per serving on average), such as those with kale and wheatgrass as the main ingredient. The frequency of use of juice made from these powders was less than 2% of the total of 1571 diet records.

Participants were offered 65 dollars for participation, except for government employees.

### 2.3. Statistical Analysis

For the analyses, we used information obtained from the 131 participants. Each participant had twelve separate one-day diet records at the end of the survey. Analyses were conducted using information on energy and 39 nutrients from 131 participants. We evaluated the reliability of the repeated measures data by use of the intraclass correlation coefficient (ICC) [19,20,21]. The ICC, the ratio of between-person variance to total variance, is estimated by analysis of variance and estimation of variance components [19]. We obtained ICCs with 95% confidence intervals to estimate reliability as follows:(1)$$\mathrm{ICC}={\sigma_{b}}^{2}/({\sigma_{w}}^{2}+{\sigma_{b}}^{2})$$
where σ_b_^2^ is the between-person variation and σ_w_^2^ is the within-person variation [20,21], from linear mixed models [4] that account for repeated measures as implemented by PROC MIXED [22].

We calculated the number of days required to estimate a person’s true intake with a specified degree of error by use of the following formula [1,2]:(2)$$n={\left(Z_{\alpha }{\mathrm{CV}}_{w}/D_{o}\right)}^{2}$$
where n = the number of days needed per person, Z_α_ = the normal deviate for the percentage of times the measured value should be within a specified limit, CV_w_ = the within-person coefficient of variation, D_o_ = the specified limit as a percentage of true long-term intake. We estimated the number of days with a 95% confidence interval within 10%, 20%, 30%, and 40% of their true intake. The CV_w_ and the ratio of within- to between-person variance (R_w/b_) were calculated to indicate the degree of within-person variation for each specific nutrient.

For energy adjustment, we used the residual model [1,3,23,24,25]. We first computed residuals from regression models [24], with total caloric intake as the independent variable and the nutrient intake values as the dependent variable. The residuals were added to the expected nutrient value for the mean caloric intake to obtain a score adjusted to the average caloric intake (1846 kcal). In the regression modeling, we used the natural logarithm of the nutrient values as dependent variables because most nutrient intakes skewed toward higher values. We log-transformed all variables after setting zeros to a small non-zero value (0.0001 unit/day) and then calculated residuals on the log scale [26,27,28]. The frequency distribution of daily alcohol intake was skewed toward higher intakes (30% of zero intakes out of 1571 diet records and median = 0.7 g/day); therefore, we estimated the ICC using Spearman’s ranked data to incorporate the large number of zero values of the dataset. The SAS software package (Ver. 9.4; SAS Institute Inc.) was used to perform all statistical analyses.

### 2.4. Ethical Considerations

The study was approved by the Research Ethics Committee of the University of Nagasaki (the project approved number 211). All subjects gave their written informed consent for inclusion before they participated in the study.

## 3. Results

The study participants were generally healthy individuals, 33 males, and 98 females, residing in Nagasaki City, Togitsu-cho Town, Saikai City, Nagayo-cho Town, Ohmura City, or Iki City (an island), aged 30 to 73 years with an average age of 58 years (54 in men and 59 in women). The mean body mass index (BMI, weight (kg)/height (m^2^)) was 23.6 ± 3.3 kg/m^2^ (24.4 ± 2.9 in men and 23.3 ± 3.3 in women). The rate of current smoking was 5% (18% in men and 1% in women). The mean intakes of selected nutrients in the current study were similar to those in the NHNS 2016 (n = 26,133; supplements not included) [17], except that intakes were higher for folate, vitamin C, niacin equivalent (NE), and potassium. Percentages of energy intake from carbohydrate, protein, and lipids were comparable between this study and the NHNS 2016 (Appendix A). In the current study, mean intakes for most of the nutrients did not relate to the sequence of the records (day 1–12) except that intakes were lower for beta-cryptoxanthin from late spring to summer in the region when mandarins are less available (Appendix A). Our sample included 220 records on Sunday, 197 on Monday, 286 on Tuesday, 263 on Wednesday, 252 on Thursday, 186 on Friday, and 167 on Saturday, and the mean intakes of all nutrient did not vary significantly by day of the week. For selected nutrients, the mean intake levels were generally similar between nutrition volunteers and government workers. The mean intake level of the selected nutrients was similar between the weekdays and the weekends.

Table 1 displays the ICCs with 95 percent confidence intervals (95% CI) for nutrient intakes with and without adjustment for total energy intake. Among the unadjusted nutrients, ICCs ranged from 0.05 (95% CI = 0.03–0.09) for vitamin A retinol equivalent (RE) to 0.55 (95% CI = 0.48–0.62) for potassium. After energy adjustment, ICCs decreased for all of the nutrients. ICC was 0.02 (95% CI = 0.01–0.07) for vitamin A (RE) and 0.52 (95% CI = 0.45–0.59) for potassium. Intakes of macronutrients tended to have moderate degrees of day-to-day variation (energy-adjusted CV_w_, range = 0.13–0.23, mean 0.17), while the degree of variation for intakes of micronutrients varied dramatically (energy-adjusted CV_w_, range = 0.17–2.59, mean = 0.54). Similarly, the ratios of within- to between-person variance (R_w/b_) tended to be smaller for macronutrients than those for vitamins and minerals. The R_w/b_ was 1.2 for protein and 1.7 for lipids before energy adjustment, while these were 2.3 for protein and 2.4 for lipids after energy adjustment. Among the micronutrients, before adjustment for energy the R_w/b_ ranged from 0.8 for potassium to 19.0 for vitamin A (RE). After adjusting for energy, the R_w/b_ was 0.9 for potassium and 49.0 for vitamin A (RE). In the analysis for alcohol by use of the Spearman’s ranked data, the ICC was 0.54 before adjustment and 0.47 after adjustment (Table 1). Table 2 presents the coefficients of within-person variation (CVw) with 95% CIs and the number of days needed to estimate a person’s true intake within errors of 10%, 20%, 30%, and 40%. Among the unadjusted nutrients, the CVw was 0.20 (95% CI = 0.19–0.21) for carbohydrate and 1.72 (95% CI = 1.53–1.92) for vitamin A (RE). After energy adjustment, it was 0.13 (95% CI = 0.12–0.13) for carbohydrate and 1.72 (95% CI = 1.53–1.88) for vitamin A (RE). Intakes of macronutrients after energy-adjustment tended to have moderate degrees of day-to-day variation (CVw range = 0.13 to 0.23, mean = 0.17), while the degree of variation for intakes of micronutrients varied dramatically (CVw range = 0.17–2.59; mean = 0.54). The CVw decreased for most of the energy-adjusted nutrients. To estimate the true level of nutrient intake within 10%, the numbers of days needed for unadjusted protein and lipid were 19 and 45, respectively, and 12 and 21 for energy-adjusted intakes (Table 2).

Appendix A shows ICCs and 95% CIs, and R_w/b_ for energy-adjusted nutrient intakes by sex. The trends in ICCs for macronutrients and micronutrients were generally similar between males and females. The within-person variance of vitamin A (RE) and beta-cryptoxanthin was larger for females than for males. Appendix A shows the sex-specific within-person variation and the number of days needed to estimate a person’s true nutrient intake within 10% of true intake after energy adjustment. The trend in coefficients of within-person variation for macronutrients and micronutrients and the number of days required to estimate a person’s true intake were similar between men and women.

## 4. Discussion

Among the 131 residents in the western region of Japan, we observed considerable day-to-day variation in nutrient intake, assessed by twelve one-day diet records over one year. Intakes of macronutrients tended to have moderate degrees of day-to-day variation, but intakes of micronutrients varied dramatically. When total energy was adjusted, the ratio of within- to between-person variation became higher for most nutrients. For within- and between-person variation for selected crude nutrients, our findings were similar to the results from most previous studies in western countries and in Japan [12,13]. For example, for total energy, the R_w/b_ ranged from 0.8 to 2.2 in western countries [1,5,6,8,9,29] and from 1.0 to 2.2 in Japan. For protein, it ranged from 1.2 to 3.9 in the western populations and from 1.8 to 2.2 in the Japanese population. In the current study, we observed an R_w/b_ of 1.0 for total energy and 1.2 for unadjusted protein. Our study showed that the R_w/b_ was greater when energy was adjusted, which was as expected [1]. For the coefficient of within-person variation, our findings were similar to the results of the study in the US [1]. We cannot compare energy-adjusted R_w/b_ and coefficients of within-person variation from our study to the values from the Japanese studies because values after energy adjustment were not reported. Also, information on ICCs by diet records or recalls in the previous studies in western countries and Japan is limited. The analyses in our study showed that the variation in total energy intake contributed to the variation in nutrient intakes; energy-adjustment has driven the ICCs towards lower values, as expected [1,3]. Similarly, it led to the lower values of the coefficient of within-person variation. As a consequence, additional days of information will be needed to distinguish between individuals in epidemiologic studies, because energy-adjusted intakes are usually considered the most important.

When total energy was adjusted, for most nutrients, when the ICC became lower, the R_w/b_ became higher because the proportion of within-person variation became substantially higher. Therefore, we presented energy-adjusted nutrient intakes, which could represent R_w/b_ and ICC after accounting for the differences in total energy intake among individuals. Showing the results for energy-adjusted intakes is important because these intakes are of primary interest in nutritional epidemiology due to the fact that a person must change their intake of specific nutrients primarily by changing the quality of the diet rather than by changing their total energy intake. Large day to day variation may not be important physiologically for nutrients that are stored in the body with a long half-life, like fat soluble vitamins. However, for the assessment of the intake of these nutrients, the day to day variation is still very important.

The inclusion of highly motivated persons as participants might tend to make the results less generalizable to the overall population. However, we do note that the average intakes are quite similar to the national and prefectural average intakes [17,30,31]. For example, the mean nutrient intakes were similar to the findings from the NHNS 2016, conducted among a stratified random sampling out of the 462 districts [32] from the National Census Survey 2010 [33]; the mean intakes were 1865 kcal for energy, 68.5 g for protein, and 57.2 g for lipids. Of note, the mean intakes of folate, vitamin C, and potassium in the current study were somewhat higher than those in the NHNS 2016 but they were comparable to the values of the elderly of the national and prefectural surveys. In general, the intakes of these nutrients among the elderly were higher than those among younger people in the national and prefectural surveys, which might reflect their higher consumption of fruits and vegetables [17,30,31,34].

A strength of the current study is the data collection method. To incorporate variation due to seasonality and day of the week, we collected dietary information from the participants for one day 12 times over a one-year period. To increase the reliability, we complemented the diet record method with the 24-hour recall method. The recruitment method of the highly motivated persons was one limitation in this study but we achieved the compliance rate by 99% through this approach.

Some potential limitations should be noted in the interpretation of our findings. First, sampling bias could have occurred in the selection of the study population. However, the study area covered multiple urban and rural areas to capture diverse geographical and cultural characteristics. Second, due to the limited specificity of foods in the Standard Tables of Food Composition Japan 2010, misclassification of food and drinks could have occurred during the coding process to calculate the nutrients intakes from the diet records.

When specific foods or beverages were not listed in the tables, we replaced those foods with foods of similar nutrient content (approximately 2%). Third, the mean nutrients intakes might have been overestimated when compared to the mean intakes of folate, vitamin C, and carotenoids in the NHENS 2016 because we included the intake of supplements of dried green vegetable powders. However, it should not have affected the mean nutrients intakes of the current study because the frequency of use was small (less than 2% of the total 1571 diet records) and one serving size was small (5 g per on average). For future work, the Standard Table of Food Composition Japan should include additional food and beverages. Even though we have not evaluated these limitations in the current study, they should not appreciably affect the interpretation of our findings. Findings from our study indicate that the short-term diet record/recall method to collect nutrition information to represent usual diet is not feasible for a large epidemiological study because we would need a large number of days for many nutrients. Thus, the use of other methods, such as food frequency questionnaires, alone or in combination with short-term methods, needs further study.

## 5. Conclusions

To our knowledge, this is the first study to report the reproducibility as intraclass correlation coefficients for crude and energy-adjusted nutrient intakes in a general Japanese population. We found considerable day-to-day variation in nutrient intakes assessed by 12 diet records over a one-year period. This study provided information that may be useful for other dietary studies in Japanese populations.

## Figures and Tables

**Table 1 nutrients-11-02515-t001:** Intraclass correlation coefficients (ICC) and the ratio of within- to between-person variance (Rw/b) for daily intakes of unadjusted- and adjusted-nutrients (data provided by 131 residents in Japan)*.

Nutrients	Mean^e^	SD^f^	Unadjusted		Adjusted^g^	
			ICC^a,b^ (95%CI^c^)	R_w/b_^d^	ICC (95%CI)	R_w/b_
Energy/kcal	1846	354	0.49 (0.42, 0.55)	1.0		
Protein/g	69.9	15.0	0.45 (0.38, 0.52)	1.2	0.30 (0.24, 0.36)	2.3
Lipid/g	56.5	15.9	0.37 (0.31, 0.44)	1.7	0.29 (0.24, 0.36)	2.4
SFA^h^/g	15.65	4.99	0.34 (0.28, 0.41)	1.9	0.31 (0.25, 0.38)	2.2
MUSFA^i^/g	20.51	6.52	0.34 (0.28, 0.41)	1.9	0.27 (0.22, 0.34)	2.7
PUFA^j^/g	12.34	3.31	0.28 (0.22, 0.35)	2.6	0.14 (0.10, 0.19)	6.1
n-3PUFA/g	2.3	0.9	0.23 (0.17, 0.29)	3.3	0.13 (0.09, 0.18)	6.7
n-6PUFA/g	9.9	2.7	0.26 (0.20, 0.32)	2.8	0.14 (0.10, 0.19)	6.1
Cholesterol/mg	322	107	0.28 (0.22, 0.34)	2.6	0.12 (0.09, 0.18)	7.3
Carbohydrate/g	247	49	0.47 (0.40, 0.54)	1.1	0.38 (0.31, 0.45)	1.6
Total fiber/g	15.4	4.8	0.50 (0.43, 0.57)	1.0	0.49 (0.42, 0.56)	1.0
Soluble fiber/g	3.4	1.2	0.38(0.32, 0.45)	1.6	0.35 (0.29, 0.42)	1.9
Insoluble fiber/g	11.3	3.5	0.50 (0.43, 0.56)	1.0	0.48 (0.41, 0.55)	1.1
Vitamin A/μg RE^k^	561	353	0.05 (0.03, 0.09)	19.0	0.02 (0.01, 0.07)	49.0
Vitamin D/μg	7.9	4.2	0.12 (0.08, 0.17)	7.3	0.05 (0.03, 0.09)	19.0
alpha-Tocopherol/mg	7.1	1.9	0.30 (0.24, 0.36)	2.3	0.25 (0.19, 0.31)	3.0
Vitamin K/μg	232	103	0.35 (0.28, 0.42)	1.9	0.33 (0.26, 0.39)	2.0
Vitamin B_1_/mg	0.94	0.21	0.21 (0.16, 0.27)	3.8	0.13 (0.09, 0.18)	6.7
Vitamin B_2_/mg	1.39	0.35	0.43 (0.37, 0.50)	1.3	0.32 (0.26, 0.39)	2.1
Niacin/mg NE^l^	30.9	7.2	0.42 (0.35, 0.49)	1.4	0.22 (0.17, 0.28)	3.5
Vitamin B_6_/mg	1.34	0.4	0.48 (0.42, 0.55)	1.1	0.40 (0.33, 0.47)	1.5
Vitamin B_12_/μg	6.5	3.3	0.14 (0.10, 0.19)	6.1	0.06 (0.04, 0.10)	15.7
Folate/μg	410	128	0.42 (0.35, 0.49)	1.4	0.35 (0.29, 0.42)	1.9
Pantothenic acid/mg	6.08	1.39	0.45 (0.39, 0.52)	1.2	0.33 (0.26, 0.39)	2.0
Vitamin C/mg	143	55	0.46 (0.39, 0.53)	1.2	0.41 (0.34, 0.48)	1.4
Sodium/mg	3624	856	0.34 (0.27, 0.40)	1.9	0.23 (0.17, 0.29)	3.3
Salt-Eq/g	9.1	2.2	0.33 (0.27, 0.40)	2.0	0.23 (0.17, 0.29)	3.3
Potassium/mg	2794	732	0.55 (0.48, 0.62)	0.8	0.52 (0.45, 0.59)	0.9
Calcium/mg	538	164	0.39 (0.33, 0.46)	1.6	0.35 (0.29, 0.42)	1.9
Magnesium/mg	278	67	0.38 (0.32, 0.45)	1.6	0.28 (0.22, 0.35)	2.6
Phosphorus/mg	1035	232	0.49 (0.42, 0.56)	1.0	0.38 (0.32, 0.45)	1.6
Iron/mg	8.7	2.3	0.44 (0.38, 0.51)	1.3	0.36 (0.30, 0.43)	1.8
Selenium/μg	62	18	0.19 (0.15, 0.25)	4.3	0.07 (0.04, 0.11)	13.3
Zinc/mg	1.2	0.3	0.33 (0.27, 0.40)	2.0	0.17 (0.13, 0.23)	4.9
Cupper/mg	4.35	1.29	0.44 (0.37, 0.51)	1.3	0.28 (0.22, 0.35)	2.6
Manganese/mg	1.29	1.62	0.53 (0.46, 0.60)	0.9	0.47 (0.40, 0.54)	1.1
beta-carotene/μg	3469	1696	0.28 (0.22, 0.35)	2.6	0.25 (0.20, 0.31)	3.0
alpha-carotene/μg	563	350	0.20 (0.15, 0.26)	4.0	0.01 (0.00, 0.08)	99.0
beta-cryptoxanthin/μg	528	443	0.13 (0.09, 0.18)	6.7	0.07 (0.04, 0.11)	13.3

* The data in this study were collected between 2014 and 2015 in Nagasaki Prefecture, Japan. The title of this study is Reliability of Repeated Measures of Nutrient Intake by Diet Records in Residents in the Western Region of Japan. ^a^ ICC: intraclass correlation coefficient. ^b^ Each of the 130 participants provided 12 diet records, and one submitted 11 records. Analysis was done using 1571 records. ^c^ CI: confidence interval. ^d^ R_w/b_: σ_w_^2^/σ_b_^2^, the ratio of within- to between person variance. ^e^ Means of the total study population. ^f^ SD: standard deviation. ^g^ Adjustment by use of the residual model method. ^h^ SFA: saturated fatty acid. ^i^ MUFA: monounsaturated fatty acid. ^j^ PUFA: polyunsaturated fatty acid. ^k^ RE: retinol equivalent. ^l^ NE: niacin equivalent.

**Table 2 nutrients-11-02515-t002:** Coefficients of within-person variation (CV_W_) and the number of days required to estimate the true intakes of unadjusted- and adjusted-nutrients to lie within specified % of true means (data provided by 131 residents in Japan).

		Unadjusted					Adjusted^b^			
Nutrient	CV_W_^a^ (95% CI)	Specified Limit as %				CV_W_ (95% CI)	Specified Limit as %			
		10%	20%	30%	40%		10%	20%	30%	40%
Energy/kcal	0.19 (0.18, 0.20)	14	4	2	1					
Protein/g	0.22 (0.21, 0.24)	19	5	3	2	0.17 (0.17, 0.18)	12	3	2	1
Lipid /g	0.34 (0.32, 0.36)	45	12	5	3	0.23 (0.22, 0.25)	21	6	3	2
SFA^c^/g	0.41 (0.39, 0.44)	65	17	8	5	0.31 (0.30, 0.33)	37	10	5	3
MSFA^d^/g	0.41 (0.39, 0.44)	65	17	8	5	0.31 (0.29, 0.32)	37	10	5	3
PUFA^e^/g	0.39 (0.37, 0.41)	59	15	7	4	0.32 (0.30, 0.33)	40	10	5	3
n-3PUFA/g	0.60 (0.56, 0.64)	139	35	16	9	0.55 (0.52, 0.59)	117	30	13	8
n-6PUFA/g	0.41 (0.39, 0.44)	65	17	8	5	0.35 (0.33, 0.36)	48	12	6	3
Carbohydrates/g	0.20 (0.19, 0.21)	16	4	2	1	0.13 (0.12, 0.13)	7	2	1	1
Cholesterol/mg	0.48 (0.45, 0.51)	89	23	10	6	0.48 (0.46, 0.51)	89	23	10	6
Total fiber/g	0.30 (0.28, 0.32)	35	9	4	3	0.27 (0.26, 0.29)	29	8	4	2
Soluble fiber/g	0.41 (0.39, 0.44)	65	17	8	5	0.39 (0.37, 0.42)	59	15	7	4
Insoluble fiber/g	0.30 (0.28, 0.32)	35	9	4	3	0.30 (0.28, 0.32)	35	9	4	3
Vitamin A/μg RE^f^	1.72 (1.53, 1.92)	1137	285	127	72	1.70 (1.53, 1.88)	1111	278	124	70
Vitamin D/μg	1.13 (1.03, 1.25)	491	123	55	31	1.31 (1.20, 1.44)	660	165	74	42
alpha-Tocopherol/mg	0.37 (0.35, 0.39)	53	14	6	4	0.32 (0.31, 0.34)	40	10	5	3
Vitamin K/μg	0.57 (0.52, 0.62)	125	32	14	8	0.57 (0.53, 0.62)	125	32	14	8
Vitamin B_1_/mg	0.37 (0.35, 0.39)	53	14	6	4	0.33 (0.31, 0.34)	42	11	5	3
Vitamin B_2_/mg	0.27 (0.26, 0.29)	29	8	4	2	0.25 (0.24, 0.26)	25	7	3	2
Niacin/mg NE^g^	0.26 (0.24, 0.27)	26	7	3	2	0.23 (0.22, 0.24)	21	6	3	2
Vitamin B_6_/mg	0.27 (0.25, 0.29)	29	8	4	2	0.24 (0.23, 0.26)	23	6	3	2
Vitamin B_12_/μg	1.03 (0.94, 1.13)	408	102	46	26	1.22 (1.12, 1.33)	572	143	64	36
Folate/μg	0.35 (0.33, 0.37)	48	12	6	3	0.34 (0.32, 0.36)	45	12	5	3
Pantothenic acid/mg	0.24 (0.23, 0.25)	23	6	3	2	0.21 (0.20, 0.22)	17	5	2	2
Vitamin C/mg	0.40 (0.37, 0.43)	62	16	7	4	0.40 (0.37, 0.43)	62	16	7	4
Sodium/mg	0.31 (0.29, 0.32)	37	10	5	3	0.29 (0.27, 0.30)	33	9	4	3
Salt-Eq/g	0.31 (0.29, 0.33)	37	10	5	3	0.29 (0.28, 0.31)	33	9	4	3
Potassium/mg	0.23 (0.21, 0.24)	21	6	3	2	0.20 (0.19, 0.21)	16	4	2	1
Calcium/mg	0.36 (0.33, 0.38)	50	13	6	4	0.33 (0.31, 0.35)	42	11	5	3
Magnesium/mg	0.29 (0.27, 0.30)	33	9	4	3	0.26 (0.25, 0.28)	26	7	3	2
Phosphorus/mg	0.22 (0.21, 0.23)	19	5	3	2	0.17 (0.16, 0.18)	12	3	2	1
Iron/mg	0.28 (0.26, 0.30)	31	8	4	2	0.25 (0.24, 0.27)	25	7	3	2
Selenium/μg	0.51 (0.48, 0.54)	100	25	12	7	0.50 (0.48, 0.53)	97	25	11	7
Zinc/mg	0.28 (0.26, 0.29)	31	8	4	2	0.23 (0.22, 0.24)	21	6	3	2
Cupper/mg	0.26 (0.25, 0.28)	26	7	3	2	0.26 (0.24, 0.27)	26	7	3	2
Manganese/mg	0.27 (0.25, 0.28)	29	8	4	2	0.26 (0.25, 0.28)	26	7	3	2
beta_carotene/μg	0.71 (0.65, 0.78)	194	49	22	13	0.71 (0.65, 0.77)	194	49	22	13
alpha_carotene/μg	1.08 (0.96, 1.20)	449	113	50	29	2.59 (2.24, 2.99)	2577	645	287	162
beta_cryptoxanthin/μg	1.72 (1.48, 1.99)	1137	285	127	72	2.24 (1.92, 2.62)	1928	482	215	121

* The data in this study were collected between 2014 and 2015 in Nagasaki Prefecture, Japan. The title of this study is Reliability of Repeated Measures of Nutrient Intake by Diet Records in Residents in the Western Region of Japan. ^a^ Analysis was done using 1571 diet records; each of the 130 participants provided 12 diet records and one submitted 11 records. ^b^ Adjustment by use of the residual model method. ^c^ Saturated fatty acid. ^d^ Monounsaturated fatty acid. ^e^ Polyunsaturated fatty acid. ^f^ Retinol equivalent. ^g^ Niacin equivalent.

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
