# Peer review of "Reliability of Repeated Measures of Nutrient Intake by Diet Records in Residents in the Western Region of Japan"

_nutrients, 2019, doi:10.3390/nu11102515_

Round 1
Reviewer 1 Report
The authors have included the reviewers’ comments and responded us individually, indicating exactly how we addressed each concern or problem and describing the changes they have made. The changes are marked in red in the manuscript and all authors have approved the revisions.
Author Response
Dear Reviewer 1,
Thank you very much for your comments on our manuscript for the first review. I uploaded the revised manuscript which incorporated comments from reviewer 2 for the second review.
Sincerely yours,
the corresponding author

Reviewer 2 Report
Congratulations to the Authors on collecting such a large number of dietary information (12 monthly interview) from as many as 132 respondents. Unfortunately in Europe participants often abandon this type of research after several months if they do not receive any gratification, e.g. dietary counseling.
The manuscript was prepared very well, with great care. The study was carried out thoroughly, without any methodical errors. The descriptions of the results and discussion are correct and legible. I do not have comment for these part of manuscript.
However, I have comments on Methods section:
1) Were the participants asked about "the leftovers" during interviews with the dietitians? As we know, "leftovers" can significantly affect the results of nutrition intake so whether were included during calculation nutrient intakes?
2) Lines 81-82 contains unnecessary repetitions about collection dietary information, Authors double inform that used 24-hour recall methods.
3) Lines 80-88: I suggest redrafting this fragment. In addition to repetitions, this fragment is quite chaotic and requires strong focus to understand how the data was collected.
Author Response
Please see the attachment.

This manuscript is a resubmission of an earlier submission. The following is a list of the peer review reports and author responses from that submission.
Round 1
Reviewer 1 Report
see attached document

Reviewer 2 Report
The aim of the study is to describe the crude and energy-adjusted ICCs, and within- and between-person components of variations for energy and 39 nutrients obtained from twelve 1-day diet records collected over one year from 131 residents of urban and rural areas in western Japan.
Introduction provides sufficient background and includes relevant references. There is a huge work behind this paper but the results are unclear.
The research design is appropriate, but sampling bias could have occurred in the selection of the study population. The participants were generally healthy individuals, 33 males, and 98 females, but they were aged 30 to 73 years with average age of 58 years.
The authors say that Supplemental Table 3 shows ICCs and 95% CIs, and Rw/b for energy-adjusted nutrient intakes by gender. The trends in ICCs for macronutrients and micronutrients were generally similar between males and females (data not shown)
Tha authors say that Supplemental Table 4 shows the gender-specific within-person variation and the number of days needed to estimate a person’s true nutrient intake within 10% of true intake after energy adjustment. The trend in coefficients of within-person variation for macronutrients and micronutrients and the number of days required to estimate a person’s true intake were similar between men and women (data not shown)
The authors say that to incorporate variation due to seasonality and day of the week, they collected dietary information from the participants for one day 12 times over a one-year period, by using the diet. There is no discussion of results related to seasonality.
Minor details
Line 93 English language and style are fine/minor spell check required
Line 156 space between words
Line 175 Table 2. CVW instead of CVW
I recommend the authors present the missing tables (Supplemental table 3 and 4)